# Expansion of Canine Heartworm in Spain

**DOI:** 10.3390/ani12101268

**Published:** 2022-05-14

**Authors:** José Alberto Montoya-Alonso, Rodrigo Morchón, Sara Nieves García-Rodríguez, Yaiza Falcón-Cordón, Noelia Costa-Rodríguez, Jorge Isidoro Matos, Iván Rodríguez Escolar, Elena Carretón

**Affiliations:** 1Internal Medicine, Faculty of Veterinary Medicine, Research Institute of Biomedical and Health Sciences (IUIBS), University of Las Palmas de Gran Canaria, 35413 Las Palmas de Gran Canaria, Spain; alberto.montoya@ulpgc.es (J.A.M.-A.); saranieves.garcia@ulpgc.es (S.N.G.-R.); yaiza.falcon@ulpgc.es (Y.F.-C.); noelia.costa@ulpgc.es (N.C.-R.); jorge.matos@ulpgc.es (J.I.M.); elena.carreton@ulpgc.es (E.C.); 2Zoonotic Infections and One Health GIR, Laboratory of Parasitology, Faculty of Pharmacy, University of Salamanca, Campus Miguel Unamuno, 37007 Salamanca, Spain; ivanrodriguez@usal.es

**Keywords:** *Dirofilaria immitis*, dogs, Spain, epidemiology, provinces, antigen tests

## Abstract

**Simple Summary:**

*Dirofilaria immitis* is a nematode parasite causing heartworm disease worldwide. Spain is one of the southern European countries where the risk of infection in domestic dogs is moderate/high. The aim of this study was to update the epidemiology of canine heartworm and analyse the results based on climate and other geo-environmental factors. For this, 9543 blood samples from dogs from all provinces, autonomous cities and islands that comprise Spain were tested using *D. immitis* antigen tests. The mean prevalence obtained was 6.47%. Prevalences exceeding 11% were located in the northwestern and southern provinces, in the Balearic and Canary Islands. The presence of positive dogs was described for the first time in provinces and islands where previously no cases of infected dogs had been reported. Considering its zoonotic character, further studies are needed to identify infected animals and to carry out national control programmes for the prevention of heartworm disease.

**Abstract:**

The climate of Spain has favourable characteristics for the development of *D. immitis* in dogs, being an endemic country. Given that vector-borne diseases are spreading rapidly through Europe, due to factors such as climate change, the expansion of vectors and the increased mobility of reservoir animals, the aim was to update the epidemiology of heartworm in dogs and analyse the results based on climate and other epidemiological and geo-environmental factors. To this aim, 9543 blood samples from dogs from all provinces and autonomous cities of Spain were analysed for the detection of antigens of *D. immitis*, obtaining a prevalence of 6.47%. The northwestern and southern provinces showed the highest prevalences, as well as in the Balearic and Canary Islands. Prevalences were higher in dogs outdoors. Furthermore, most of the positive dogs were found in regions with high humidity and water availability. This study shows, for the first time, positive cases in provinces and islands where no cases had previously been described and demonstrates the continuous expansion and consolidation of heartworm in Spain. Considering its zoonotic character, the implementation of control and awareness programmes for the prevention of *D. immitis* in pets is necessary.

## 1. Introduction

Heartworm disease, caused by *Dirofilaria immitis*, is a cosmopolitan disease that affects canids and felids, both domestic and wild, worldwide. The domestic dog is the main reservoir and the animal in which the epidemiology of *D. immitis* has been studied the most [1]. In this host, *D. immitis* causes a chronic pathology that mainly affects lungs and right cardiac chambers, caused by the presence of adult worms and the release of the endosymbiotic bacteria *Wolbachia pipientis*. It is a severe disease; the chronic presence and/or sudden death of the adult worms can cause congestive heart failure and the death of the infected animal [2,3]. However, heartworm infection is mainly asymptomatic, and symptoms are generally only shown when the disease caused by the presence of the parasite is advanced and there is marked vascular damage. Clinical signs commonly displayed by the infected dogs are chronic cough, dyspnoea, weakness, exercise intolerance, syncope, anorexia, weight loss, ascites or abnormal cardiac and heart sounds [4].

It is a vector-borne zoonotic disease in which culicid mosquitoes of the genera *Culex* spp., *Aedes* spp. and *Anopheles* spp., among others, act as vectors. Since the presence and proliferation of these vectors is directly influenced by climatic factors, mainly humidity and temperature, the distribution of heartworm is directly determined by the climate; in this sense, climate change caused by the global warming is possibly facilitating the spread of vector-borne diseases [1]. Moreover, the influence of other factors in its distribution is also acknowledged, as seen by the global increase in the mobility of reservoir animals from endemic areas, or environmental changes caused by human activity such as agricultural crops or urbanisations [5].

Heartworm disease is an endemic disease that is continuously expanding in the European continent from the southern countries, traditionally endemic, to the centre and north of the continent, while some studies have also noted the presence of stable or declining prevalences in hyperendemic areas, probably due to the solid establishment of prophylaxis programmes [1,2,5,6]. In Spain, there are different studies that have evaluated the epidemiological status of the disease in dogs, by the determination of circulating antigens or microfilariae, or by the presence of adult worms at necropsy [1,7,8,9]. In addition, prediction models of the infection have been made [10,11]. The prevalences previously reported showed a high risk of infection throughout the Spanish territory, with the exception of some central and northern autonomous communities [9]. However, prevalences in a same region can vary notably depending on the orography and climate [1,7,8,9,10,11,12,13]. Furthermore, larvae of *D. immitis* in *Cx. pipiens* in the Iberian Peninsula [14,15] and in *Cx. theileri* in the Canary Islands [16] have been found.

These studies mentioned above demonstrate a continued and worrying expansion of canine heartworm disease in Spain. For this reason, it is necessary to carry out ongoing epidemiological studies in this country, aimed to monitor the progress of *D. immitis*. Therefore, the objective of this study was to update the epidemiology of canine heartworm and analyse the results based on climate and other epidemiological and geo-environmental factors.

## 2. Materials and Methods

### 2.1. Location and Climatology

Spain is located in the south of Western Europe. The country occupies most of the Iberian Peninsula, the Balearic Islands which are located in the western Mediterranean Sea between Spain and Italy, the Canary Islands which are positioned in the northeastern Atlantic Ocean, as well as Ceuta and Melilla, which are autonomous cities placed in the north of the African continent. Spain has an area of 505,370 km², being the fourth largest country on the European continent. It is one of the most mountainous countries in Europe, having an average altitude of 650 metres above sea level. It is distributed in 50 provinces, belonging to 17 autonomous communities and 2 autonomous cities (Ceuta and Melilla) (Figure 1).

In Spain, seven climates predominate according to the Köppen classification [17,18]. The hot-summer Mediterranean climate (Csa) predominates in the Iberian Peninsula and the Balearic Islands, which is characterised by hot summers with mean temperatures in the warmest month >22 °C, which prevail in the provinces located in the southern half of the Peninsula and Mediterranean coast, except some provinces in the southeast coast, and in Balearic Islands as well. In the north coast and provinces near the Pyrenees, the temperate oceanic climate (Cfb) predominates, with temperate summers with an average temperature of the warmest month below 22 °C and an average temperature above 10 °C > 4 months of the year. In the northeast, the prevailing climates are Cfb and humid subtropical climate (Cfa); the latter is characterised by humid or sub-humid summers and no significant precipitation difference between seasons. In the northwest, the warm-summer Mediterranean climate (Csb) predominates, characterised by >4 months with temperature average ≥10 °C, and all months with average temperatures <22 °C. In some regions of the east, such as Murcia, Almería and Alicante, the cold semi-arid climate (Bsk) prevails, showing an average annual temperature <18 °C, rainfall between 200–500 mm and evaporation exceeding precipitation on average, but less than potential evaporation. In these regions, the hot semi-arid climates (BSh) and the hot desert climate (BWh) are also located. The BSh is a dry climate with an average annual temperature above 18 °C and a moderate rainy season, showing intense but scarce rainfall. In the BWh climate, the average annual temperature is above 18 °C, summers are hot or very hot and rainfall is very scarce. The Canary Islands show a very varied climate depending on each island. In general, the eastern islands are dominated by the BWh climate, while the western islands show a predominance of Csb and Csa climates in the north, and BWh and BSh in the south.

### 2.2. Samples and Assays

The study included 9543 blood samples randomly collected from domestic dogs presented for periodic health exams to 308 veterinary clinics and hospitals sited in one of the 19 autonomous communities and cities, between September 2018 and February 2022. The participation of the veterinary centres was voluntary while the dog owners were informed and consented to participate in the study. The criteria for inclusion of dogs were being >6 months of age, not having previous history of infection by *D. immitis* and not receiving regular chemoprophylaxis. Epidemiological data (age, sex, habitat, city and postcode) were also recorded.

Blood samples were collected from the cephalic or jugular vein, placed in 3 mL serum tubes and centrifuged. Serum samples were kept at −20 °C until tests were performed. All samples were tested for the detection of *D. immitis* antigens following immunochromatography technique (Uranotest^®^ Dirofilaria, Uranovet, Barcelona, Spain) following instructions from the manufacturer. Briefly, 20 μL of serum or plasma was added along with two drops of reagent to each of the test strips. The sensitivity of the tests declared from the manufacturer was 94% and the specificity was 100% (vs. necropsy).

### 2.3. Geographic Information System (GIS) Mapping

A map of the sampled area was constructed using ArcMap v.10.8 (ESRI, 2020 Redlands, CA, USA), including the following layers of environmental information that have been considered relevant for the dynamics of the analysed organisms and their transmission vectors: climate, rivers, lakes, lagoons, irrigated croplands and artificial and natural stagnant water. Particular symbols were added to facilitate the interpretation of the map. The samples were georeferenced by the postal code provided by the animal owner. Thus, the points shown on the maps correspond to the centroids of the polygons of the postal codes where the analyses of the dogs were carried out. Inferences were drawn from a proximity analysis between the presence points and the environmental characteristics of the layers used. The coordinate system used was gcs_ETRS_1989 and gcs_WGS_1984.

### 2.4. Statistical Analysis

Data were analysed using SPSS Base 20.0 software (SPSS Inc./IBM, Chicago, IL, USA). Descriptive analysis of the qualitative variables was carried out considering the number of cases and percentages. Chi-square and Fisher exact tests to compare proportions were performed. Sex, age, habitat, climate and the presence of *D. immitis* were considered as variables in the analysis. The significance level was established at *p* < 0.05.

## 3. Results

The presence of circulating *D. immitis* antigens was observed in 6.47% of the samples tested (Table 1). Table 1 and Figure 2 show the results obtained by provinces and autonomous communities. The provinces with the highest prevalences were Tenerife (17.32%), Ibiza (17.09%), Gran Canaria (16.03%) and La Palma (15.65%), followed by Cádiz (13.68%), Pontevedra (12.61%), La Gomera (11.54%), Mallorca (11.24%) and Huelva (11.11%). In general, prevalences <5% were observed in the provinces of the north of the peninsula as well as in the provinces of the centre-east and southeast of the peninsula, while the provinces of the centre-west, southwest and Mediterranean coast presented prevalences between 5–10%. The Canary and Balearic Islands presented prevalences above 10%, although great variations were observed in the results depending on each island. The presence of dogs positive to the antigen test was observed in all the islands and provinces studied, except on the island of El Hierro (Canary Islands).

**Table 1 animals-12-01268-t001:** Prevalences for *D. immitis* in domestic dogs in Spain by autonomous cities/provinces and climates (Köppen Climate Classification System). *Abbreviations*: *n* = number of dogs sampled; + = number of positive dogs; % = percentage of positive dogs. Legend: Csa: hot-summer Mediterranean climate; Cfb: temperate oceanic climate; Cfa: humid subtropical climate; Csb: warm-summer Mediterranean climate; Bsk: cold semi-arid climate; BSh: hot semi-arid climate; BWh: hot desert climate. (*) Note: the regions under Canary Islands and Balearic Islands do not correspond to provinces, but to islands. Ceuta and Melilla are autonomous cities, not part of any autonomous community.

Autonomous Community					Autonomous Community				
**Province**	**Climate**	** *n* **	**+**	**%**	**Province**	**Climate**	** *n* **	**+**	**%**
**Galicia**		**559**	**42**	**7.51**	29. Soria	Csb/Cfb	143	1	0.70
1. A Coruña	Cfb/Csb	187	16	8.56	30. Segovia	Csa/Csb	280	16	5.71
2. Lugo	Cfb/Csb	154	8	5.19	31. Ávila	Csa/Csb	143	6	4.20
3. Ourense	Csb	107	4	3.74	**Madrid**		**647**	**17**	**2.63**
4. Pontevedra	Cfb/Csb	111	14	12.61	32. Madrid	Bsk/Csa/Csb	647	17	2.63
**Asturias**		**152**	**3**	**1.97**	**Extremadura**		**250**	**22**	**8.80**
5. Asturias	Cfb	152	3	1.97	33. Cáceres	Csa	163	15	9.20
**Cantabria**		**161**	**3**	**1.86**	34. Badajoz	Bsk/Csa	87	7	8.05
6. Santander	Cfb	161	3	1.86	**Castilla-La Mancha**		**523**	**22**	**4.21**
**Basque Country**		**294**	**5**	**1.70**	35. Toledo	Bsk/Csa	135	10	7.41
7. Araba	Cfb	74	1	1.35	36. Guadalajara	Csa/Csb	104	5	4.81
8. Bizkaia	Cfb	106	2	1.89	37. Cuenca	Csa	74	2	2.70
9. Gipuzkoa	Cfb	114	2	1.75	38. Ciudad Real	Bsk/Csa	129	3	2.33
**Navarra**		**147**	**5**	**3.40**	39. Albacete	Bsk/Csa	81	2	2.47
10. Navarra	Cfa/Cfb/Bsk	147	5	3.40	**Andalusia**		**1154**	**86**	**7.45**
**La Rioja**		**164**	**12**	**7.32**	40. Huelva	Csa	108	12	11.11
11. La Rioja	Cfa/Cfb	164	12	7.32	41. Sevilla	Csa	305	29	9.51
**Aragon**		**366**	**19**	**5.19**	42. Cádiz	Csa	95	13	13.68
12. Huesca	Ds/Cfa/Cfb/Bsk	101	3	2.97	43. Córdoba	Csa	183	12	6.56
13. Zaragoza	Cfa/Cfb/Bsk	177	12	6.78	44. Málaga	Bsk/Csa/Csb/Df	158	12	7.59
14. Teruel	Cfb/Csb/Bsk	88	4	4.55	45. Jaén	BSh/Csa/Csb	81	1	1.23
**Catalonia**		**768**	**36**	**4.69**	46. Granada	Bsk/Csa/Csb/Df	116	4	3.45
15. Lleida	Ds/Cfa/Cfb/Bsk/Csa	111	4	3.60	47. Almería	BSh/Bsk/BW/Csa/Csb	108	3	2.78
16. Girona	Csa/Cfa/Cfb	105	2	1.90	**Canary Islands ***		**967**	**112**	**11.58**
17. Barcelona	Csa/Cfa/Cfb	385	17	4.42	48. La Palma	BSh/Csa/Csb	115	18	15.65
18. Tarragona	Csa/Bsk	167	13	7.78	49. El Hierro	BW/BSh/Csa/Csb	67	0	0.00
**Valencian Community**		**771**	**51**	**6.61**	50. La Gomera	BW/BSh/Csa/Csb	78	9	11.54
19. Castellón	Csa/Csb	94	6	6.38	51. Tenerife	BW/BSh/Csa/Csb	254	44	17.32
20. Valencia	Csa/Bsk/BSh	375	28	7.47	52. Gran Canaria	BW/BSh/Csa/Csb	237	38	16.03
21. Alicante	Csa/Csb/BSh	302	17	5.63	53. Fuerteventura	BW/BSh	115	2	1.74
**Murcia**		**264**	**26**	**9.85**	54. Lanzarote	BW/BSh	101	1	0.99
22. Murcia	Csa/Csb/BSh/BW	264	26	9.85	**Balearic Islands ***		**414**	**45**	**10.87**
**Castilla y León**		**1831**	**108**	**5.90**	55. Formentera	Bk/Csa	27	3	11.11
23. León	Df/Csb	235	8	3.40	56. Ibiza	BSh/Bsk/Csa	117	20	17.09
24. Zamora	Csa/Csb/Bsk	140	8	5.71	57. Mallorca	Bsk/Csa/Csb	169	19	11.24
25. Salamanca	Csa/Csb	258	18	6.98	58. Menorca	Csa	101	3	2.97
26. Valladolid	Csa/Csb/Bsk	251	22	8.76	**Autonomous cities ***				
27. Palencia	Csb	134	11	8.21	59. Ceuta	Csa	58	1	1.72
28. Burgos	Csb/Cfb	247	18	7.29	60. Melilla	Csa	53	2	3.77
					**TOTAL**		**9543**	**617**	**6.47**

When prevalences were analysed based on climate (Table 2, Figure 3D), the results showed the highest prevalences in the provinces where the Csa and Cfb climates predominate. Significant differences in prevalences between climates were found (χ^2^ = 14.67, df = 6, *p* < 0.0230), with significant differences being observed between BWh and Csb climates (*p* = 0.0252) and between Csa and Csb climates (*p* = 0.0007).

Table 2 shows the results according to age, sex and habitat, also showing the results obtained in each of the climates. Of the studied dogs, 52.1% were female and 47.9% were male. No significant differences were found by sex.

The age of the studied dogs went from 6 months to 17 years old, and for a better analysis were grouped into five age groups: dogs <1 year (3.12%), dogs 1–4 years (34.62%), dogs 5–10 years (48.41%), dogs 11–15 years (10.98%), and dogs >15 years (2.86%). Prevalence increased proportionally to age, and significant differences were found between age groups (χ^2^ = 34.32, df = 4, *p* < 0.0001), significant differences being observed between dogs <1 year and 5–10 years old (*p* = 0.0053), between dogs 1–4 years and 5–10 years old (*p* < 0.0001), and between dogs 1–4 years and 11–15 years old (*p* < 0.0001).

Regarding habitat, 22.8% of the dogs were indoors (dogs always kept inside the house), 44.77% were outdoors (always kept outside the house) and 32.43% were indoors/outdoors (dogs that spent at least 1–50% of their time outdoors). Significant differences in overall prevalence between habitats were found (χ^2^ = 107.9, df = 2, *p* < 0.0001), significant differences being found between outdoor and indoor dogs (*p* < 0.0001), between outdoor and indoor/outdoor dogs (*p* < 0.0001), and between indoor and indoor/outdoor dogs (*p* = 0.0001), showing that outdoor dogs are significantly at greater risk of infection.

Finally, considering the geospatial location of the infected dogs (Figure 3A–C), 96.76% were located in areas with high edaphic availability of water, such as stagnant water, irrigated agriculture, or river banks, or close to parks and green areas (<1.5 km). The remaining 3.24% of the infected animals had spent time in or near areas with these characteristics.

## 4. Discussion

Many studies have highlighted the existence of *D. immitis* in dogs in Spain, although considering the extension of the country, these have been scarce and mostly localised. The present study demonstrates the stable presence of canine heartworm in Spain, as well as reaffirms the continued expansion of *D. immitis*, also reporting infected dogs for the first time in the Basque Country, as well as in the autonomous cities of Ceuta and Melilla.

Regarding recent studies carried out by autonomous communities [9,19] and at the provincial/local level [7,11,12,13,20,21,22], in the last 15 years a slight increase in heartworm can be confirmed in most of the autonomous communities, with decreasing prevalences or with similar values only in Aragon, Catalonia, Valencian Community, Madrid and Castilla y León. In most cases, the variations in the prevalences are relatively slight, and it would be more precise to have the confidence intervals of all the cited studies, although these have not been published.

Among the autonomous communities in which an increase in the prevalence has been observed, those located in the north of the country stand out. In some of these communities (Cantabria, Asturias and Navarra), the presence of heartworm was recently described for the first time [9,22] so this study not only confirms the presence and expansion of this parasite in these regions, but also reports for the first time the presence of infected dogs in the Basque Country. Likewise, in Galicia, the trend towards the increase and consolidation of the presence of *D. immitis* continues throughout the autonomous community, especially in A Coruña, where the prevalence has increased in the last 13 years from 5.04% to the current 8.56% [20]. The expansion and consolidation are probably due to the climatic change, a greater mobility of reservoir animals, as well as the modification of the landscape caused by human beings. In turn, all this favours the presence of vector mosquitoes, especially invasive mosquito species, the presence of which is worryingly growing throughout the country [1,2,5,6,10]. This, together with the scarce knowledge about this parasite in these areas, which until recently were considered disease-free, favours the establishment of heartworm. Therefore, the results obtained in this study demonstrate the extreme importance of initiating awareness campaigns among veterinarians and owners of these areas aimed to control the expansion of *D. immitis*.

On the other hand, the results show that the prevalences remain stable, or that even slight decreases are observed in the areas of the peninsular northeast (Aragon, Catalonia) [9,10,11,23], while the communities traditionally considered endemic, such as the Mediterranean coast and the peninsular south, show slight–moderate increases in prevalence with respect to previous studies with the sole exception of the Valencian Community [7,9,23,24,25]. In all cases, these are regions in which a higher risk of infection has previously been reported [10]. This is especially striking in some provinces, such as Murcia, where the incidence continues to increase notably, rising almost two percentage points compared to the last study [9]. Murcia has optimal climatic conditions for the development of generations of mosquitoes, which is to have a temperature above 14 °C. Murcia presents these conditions from March to November, which is the longest period in all of Europe, being one of the Spanish regions with the highest risk of infection [10,26]. If we add to this the large number of areas with natural or artificial stagnant water, as well as irrigated areas, if control measures are not taken, the prevalences could continue to increase. The increase observed in other provinces, such as Seville [9,24,25], is also striking. However, in general, the increases observed are not notable and are probably due to the control measures carried out by veterinarians and owners.

The communities of the peninsular interior, such as Madrid, Castilla y León or Castilla-La Mancha show slight variations, maintaining similar prevalences to previous studies [8,9,13,19].

The results obtained in the Balearic Islands also stand out, where the prevalence has increased by more than four percentage points, currently presenting the highest prevalence after the Canary Islands. Being islands where the study of the incidence of this parasitosis is recent [9], it is possible that there was not optimal knowledge about the high incidence and there was a lack of control measures, so its prevalence could have increased notably.

In the Canary Islands, where the risk of infection is very high [10], the prevalences have remained similar or with slight oscillations during the last 20 years [9,12,27,28,29], although the presence of the parasite is reported for the first time in Lanzarote. These data suggest that, although the climatic conditions are conducive to the development of vectors on each of the islands and their high risk of infection, the use of prophylactic measures and the awareness of the owners are effective measures in controlling the incidence of heartworm disease. The presence of mosquito vectors and the introduction of new species of competent vectors must be considered as well [1,16].

The zoonotic profile of this disease is that of a zoonosis in which humans have no epidemiological role but are at risk of developing pulmonary dirofilariasis, which, although it is usually benign and asymptomatic, can be confused with a lung tumour. This risk is evidenced by studies that have shown a high presence of antibodies against *D. immitis* in inhabitants of endemic areas, such as La Rioja (11.6%), Salamanca (22%) and the Canary Islands (6.4%) [28,30,31,32,33]. Furthermore, human cases have been reported in which the presence of a pulmonary nodule has been evidenced [34,35]. Therefore, the expansion of this disease is indicative of the general increasing human risk not only for pulmonary dirofilariasis, but for all vector-borne diseases.

Both age and habitat showed the existence of significant differences, similar to that reported in other studies [8,9,11,12,13,19]. Higher positive rates were found in dogs 5–15 years, probably due to the longer exposure time to the mosquito vector. The prevalence was lower in dogs <1 year, likely due to a minor exposure to vectors, as stated in previous research [8,9,19,22]. In any case, the prevalence in this group is not non-existent, which reminds of the need to start with preventive measures from an early age.

Regarding habitat, dogs living indoors showed significantly lower prevalences, as shown in other similar studies [8,9,11,12,13,19]. As has been pointed out in previous studies, animals that live indoors are less exposed to the presence of vector mosquitoes and, thus, prevalences are lower. However, the presence of positive animals in indoor animals demonstrates the need for chemoprophylaxis, regardless of the dog’s habitat, since mosquitoes can be found inside homes and because animals generally go for a walk in parks and green areas where the presence of mosquitoes is more likely. Furthermore, the appearance and expansion of new species of invasive mosquitoes with the capacity to act as vectors, and which present diurnal feeding habits, further increases the risk of infection in these animals [36,37]. The establishment of vector populations in a given area is linked to their ability to survive and disperse, influenced by different factors such as ecological interactions, changing photoperiod, gradual decreases in temperature, biogeography, life history and physiology, among others. Among the mosquito species that best adapt to these kinds of changes are *Aedes albopictus* and *Cx. pipiens*, which are recognised vectors of heartworm and, indeed, the publications in this regard point to their geographical expansion and a greater incidence [14,15,37,38,39,40].

From a geospatial point of view (Figure 3A–C), 93.21% of infected animals were located in areas with high edaphic availability of water, as either stagnant water, irrigated agriculture or river banks and proximity to parks and green urban areas (<1.5 km). These areas provide a suitable environment for mosquito breeding, facilitating the transmission of heartworm disease [19,41]. This spatial distribution of the positive cases presents a clear association with different geo-environmental factors, such as the number of expected annual generations of mosquitoes (linked to temperature), the existence of irrigated areas and rivers in valleys protected from the winds, and the proximity to the coast, all of which are considered as risk factors for *D. immitis* transmission [10]. All the municipalities in which infected animals were found are located in areas with these characteristics. The results confirm that all the provinces present risk of infection, if we take into account the geolocation of the infected dogs, as well as the weather and the number of samples analysed.

## 5. Conclusions

This study demonstrates the continued expansion and consolidation of heartworm in Spain, including the northern regions, with positive cases reported for the first time in provinces and islands where previously no cases had been described. In general, the reported prevalences are higher than those described in previous studies, which emphasises the need to carry out specific programmes for early detection of heartworm in dogs, analysis of risk factors from a *One Health* point of view, identification of vectors transmitting the disease and the implementation of prevention and awareness campaigns aimed to control the expansion of the disease in pets.

## Figures and Tables

**Figure 1 animals-12-01268-f001:**
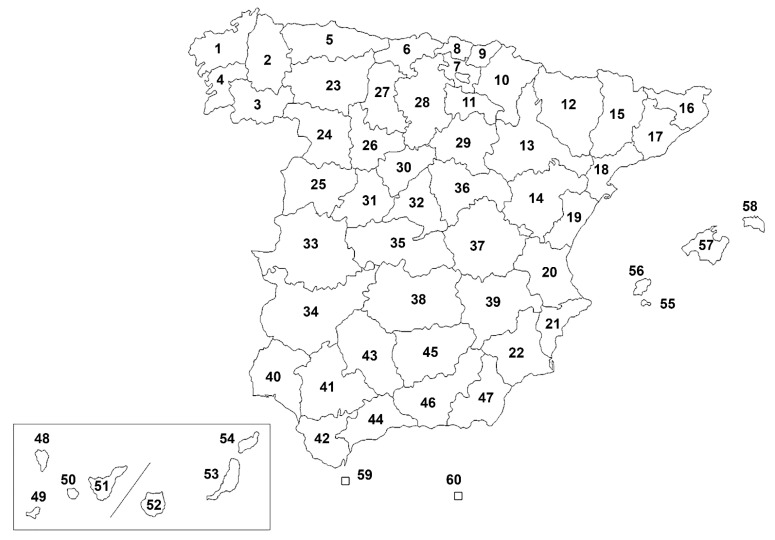
Provinces, islands and autonomous cities of Spain tested in this study. The numbers correspond to the provinces, islands and autonomous cities described in Table 1.

**Figure 2 animals-12-01268-f002:**
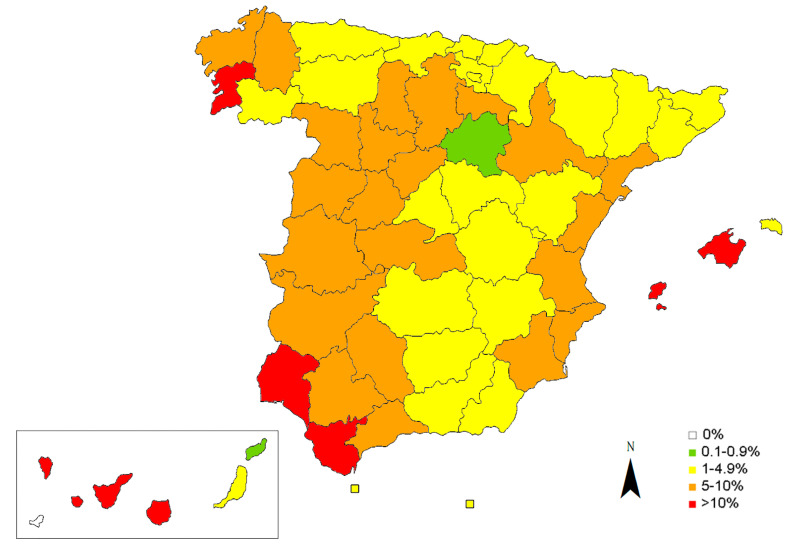
Prevalence map for *D. immitis* in domestic dogs in Spain by provinces, islands and the autonomous cities.

**Figure 3 animals-12-01268-f003:**
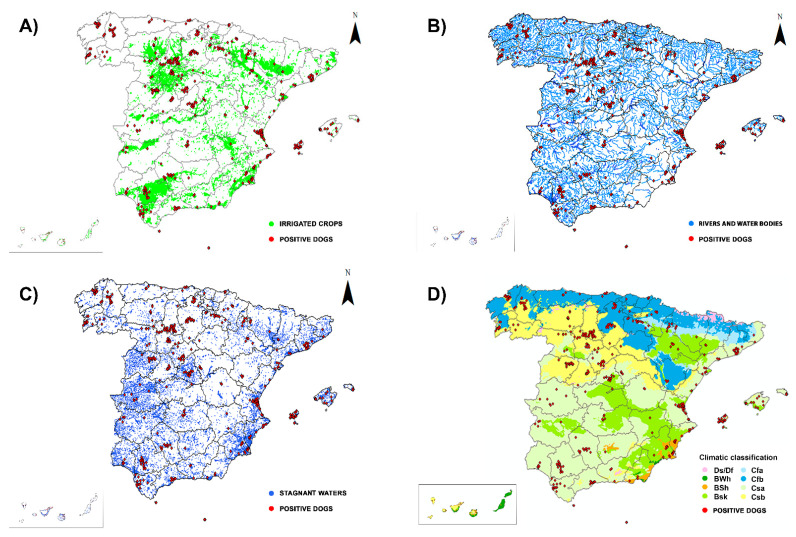
Location of irrigated groups (**A**), rivers and water bodies (**B**), artificial and natural stagnant waters (**C**) and climates (Köppen Climate Classification System) (**D**) in Spain and geolocation of the dogs infected by *Dirofilaria immitis*.

**Table 2 animals-12-01268-t002:** Distribution of prevalences for *D. immitis* in domestic dogs in Spain by climates (Köppen Climate Classification System) according to sex, age and habitat. *Abbreviations*: *n* = number of dogs sampled; + = number of positive dogs; % = percentage of positive dogs. Legend: Csa: hot-summer Mediterranean climate; Cfb: temperate oceanic climate; Cfa: humid subtropical climate; Csb: warm-summer Mediterranean climate; Bsk: cold semi-arid climate; BSh: hot semi-arid climate; BWh: hot desert climate.

		BSh			Bsk			BWh			Cfa			Cfb			Csa			Csb			Total		
		*n*	+	%	*n*	+	%	*n*	+	%	*n*	+	%	*n*	+	%	*n*	+	%	*n*	+	%	*n*	+	%
**Sex**	**Female**	157	10	6.37	234	14	5.98	196	9	4.59	67	4	5.97	336	24	7.14	2678	153	5.71	1304	97	7.44	**4972**	**311**	**6.26**
	**Male**	145	10	6.90	203	11	5.42	168	8	4.76	57	4	7.02	401	27	6.73	2451	148	6.04	1146	98	8.55	**4571**	**306**	**6.69**
**Age**	**<1 year**	5	1	20.00	12	1	8.33	15	0	0.00	1	0	0.00	21	1	4.76	193	5	2.59	51	3	5.88	**298**	**11**	**3.69**
	**1–4 years**	72	5	6.94	135	5	3.70	104	4	3.85	16	1	6.25	198	14	7.07	1994	87	4.36	785	42	5.35	**3304**	**158**	**4.78**
	**5–10 years**	113	10	8.85	167	12	7.19	173	7	4.05	71	6	8.45	391	29	7.42	2443	162	6.63	1262	121	9.59	**4620**	**347**	**7.51**
	**11–15 years**	109	4	3.67	109	6	5.50	69	5	7.25	35	1	2.86	89	5	5.62	316	39	12.34	321	27	8.41	**1048**	**87**	**8.30**
	**>15 years**	3	0	0.00	14	1	7.14	3	1	33.33	1	0	0.00	38	2	5.26	183	8	4.37	31	2	6.45	**273**	**14**	**5.13**
**Habitat**	**Outdoors**	185	16	8.65	269	21	7.81	154	10	6.49	92	7	7.61	308	27	8.77	1985	163	8.21	1244	129	10.37	**4237**	**373**	**8.80**
	**Indoors**	94	2	2.13	157	3	1.91	46	1	2.17	21	0	0.00	185	6	3.24	1281	22	1.72	391	11	2.81	**2175**	**45**	**2.07**
	**Indoors/Outdoors**	23	2	8.70	11	1	9.09	164	6	3.66	11	1	9.09	244	18	7.38	1863	116	6.23	815	55	6.75	**3131**	**199**	**6.36**
		**302**	**20**	**6.62**	**437**	**25**	**5.72**	**364**	**17**	**4.67**	**124**	**8**	**6.45**	**737**	**51**	**6.92**	**5129**	**301**	**5.87**	**2450**	**195**	**7.96**	**9543**	**617**	**6.47**

## Data Availability

The data presented in this study are available on request from the corresponding author.

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
