# Peer review of "Expansion of Canine Heartworm in Spain"

_animals, 2022, doi:10.3390/ani12101268_

Round 1
Reviewer 1 Report
The study is based on extensive sampling and the statistical analysis is appropriate, so in my opinion the paper is worthy for publication.
Anyway, some minor revisions are suggested. In detail:
Number of pages: the page number after page n. 6 and table 2 restarts from 1
Line 61: in general, heartworm disease is expanding in European continent from southern to northern countries, but in certain areas an inverse trend has been observed; e.g. in Italy, the northern part of the country is traditionally endemic and the incidence is increasing in the South.
Line 79: occupies, not occupy
Line 102: “< 22 °C” spacing to be corrected “<22°C”
Line 114-116: the number of collected samples is remarkable and quite surprising if considering that the samples were randomly collected among dogs presented for routine health examination and not receiving regular prophylaxis. The dogs population presented at a routine clinical visit represents a possible bias, as these are animals that the owners take care of; it is conceivable that there is a wider dog population that has never been led to a veterinary visit. I am asking if the Authors could give an idea of the general percentage of dogs subjected to prophylaxis. Is the prophylaxis so uncommon?
Line 126: 20 microliters of serum or plasma were added
Line 127: I suggest changing “The sensitivity of the test was” to “The sensititvity of the test declared from the manufacturer was”
Table 1 and 2: to improve the data interpretation, please report the legenda of the climate areas
Line 179-185: I suggest completing the results regarding age class stating that the prevalence increases proportionally to age
Line 186-192: as well, I suggest completing these results stating that the outdoor dogs are significantly at greater risk
Line 198: please reconsider the verb (and its conjugation) “evidence”, maybe “highlighted” could be better
Line 216-217 (and following prevalences): to estimate the increased prevalence, the confidence intervals of the estimates should be considered
Line 234-235: I cannot understand the meaning of “highest period”, could the Authors better explain its meaning?
Line 258: I suggest considering that the zoonotic profile of the disease is that of a rare zoonosis in which humans have no epidemiological role. Maybe it may be worth pointing out that the expansion of this disease is indicative of the general increasing human risk for all vector-borne diseases.
Line 275-277: maybe the Authors may also consider the growing impact of mosquitoes with diurnal behaviour
Author Response
All changes suggested by the reviewer have been added to the text and marked in red. In addition, the article has been reviewed by a proofreading /editing service. The authors thank the reviewers for their dedication and time spent reviewing this article.
REVIEWER 1. In the attached file you will find our response to your suggestions, which have been taken into account.
Thank you for enriching the manuscript.

Reviewer 2 Report
The manuscript entitled "Expansion of canine heartworm in Spain" aims to update the epidemiology of canine heartworm and analyzes the results according to climate and other geo-environmental factors. The manuscript is well written, and should be of great interest to the readers.
I recommend some editing:
-We recommend replacing Table 2 with combined histogram charts to make the results more immediate
-Overall, the paper is well written and highlights how climate change plays a role in the spread of Dirofilaria. However, an entomological aspect is missing in the analysis performed by the authors. I consider it necessary to integrate the entomological bibliographic data before accepting the manuscript in this journal.
Author Response
All changes suggested by the reviewer have been added to the text and marked in red. In addition, the article has been reviewed by a proofreading /editing service. The authors thank the reviewers for their dedication and time spent reviewing this article.
REVIEWER 2. In the attached file you will find our response to your suggestions, which have been taken into account.
Thank you for enriching the manuscript.

Reviewer 3 Report
Do the results of this study include the results of the previous study (DOI: https://doi.org/10.3389/fvets.2020.564429)? If yes, I'm not sure if this is proper. I think that the study should include a period between February 2020 and February 2022, and then compare the results of this study with the results of that previous research. The previous 2-year study showed average prevalence 6.25 %, and this study showed the prevalence 6.47 %. Thus the prevalence increases. However, what was the prevalence between Feb. 2020 and Feb. 2022?
It seems strange that there are different sources of funding of partially probably the same studies (this study and that study DOI: https://doi.org/10.3389/fvets.2020.564429).
Moreover, if the results of that previous research are included to this study, why is there a lower number of D. immitis cases in current research from Catalonia (36 cases) than the number in that previous study (39 cases)?
Are the Uranotest Quattro and Uranotest Dirofilaria the same tests? It seems they are the same tests: in the previous article and in this manuscript "serum or plasma was added along with two drops of reagent to each of the test strips".
I would be very strange if this study does not partially cover the previous research (DOI: https://doi.org/10.3389/fvets.2020.564429). All the authors of that previous research are the authors of this manuscript.
According to table 1 six hundred forty seven blood samples were collected, and D. immitis was detected in 647 samples. It means 100 %.
Do the results in this study also include the results of the other previous research (DOI: https://doi.org/10.3389/fvets.2021.784331)?
I did not find a word that this research is a continuation of the previous studies, and include part of the previous results.
Am I wrong? And the same authors collected at the same time (Sep. 2018 to Feb. 2020) in the same locations some other blood samples? And did not examine them that time?
Author Response
All changes suggested by the reviewer have been added to the text and marked in red. In addition, the article has been reviewed by a proofreading /editing service. The authors thank the reviewers for their dedication and time spent reviewing this article.
Reviewer 3:
Do the results of this study include the results of the previous study (DOI: https://doi.org/10.3389/fvets.2020.564429)? If yes, I'm not sure if this is proper. I think that the study should include a period between February 2020 and February 2022, and then compare the results of this study with the results of that previous research. The previous 2-year study showed average prevalence 6.25 %, and this study showed the prevalence 6.47 %. Thus the prevalence increases. However, what was the prevalence between Feb. 2020 and Feb. 2022?
It seems strange that there are different sources of funding of partially probably the same studies (this study and that study DOI: https://doi.org/10.3389/fvets.2020.564429).
Moreover, if the results of that previous research are included to this study, why is there a lower number of D. immitis cases in current research from Catalonia (36 cases) than the number in that previous study (39 cases)?
Are the Uranotest Quattro and Uranotest Dirofilaria the same tests? It seems they are the same tests: in the previous article and in this manuscript "serum or plasma was added along with two drops of reagent to each of the test strips".
I would be very strange if this study does not partially cover the previous research (DOI: https://doi.org/10.3389/fvets.2020.564429). All the authors of that previous research are the authors of this manuscript.
Do the results in this study also include the results of the other previous research (DOI: https://doi.org/10.3389/fvets.2021.784331)?
I did not find a word that this research is a continuation of the previous studies, and include part of the previous results.
Am I wrong? And the same authors collected at the same time (Sep. 2018 to Feb. 2020) in the same locations some other blood samples? And did not examine them that time?
Dear reviewer. First of all, we appreciate the time spent reviewing this study, as well as those previously written by our research group. Indeed, one of our lines of research is oriented towards epidemiological studies of vector-borne diseases, mainly heartworm in animals and humans. We have been publishing in this field for many years, thus accumulating a certain prestige among the scientific community and clinical veterinarians. This allows us to have a solid network of collaborations with veterinarians who work in clinics, hospitals and animal shelters, which has allowed us to have a large sera collection in our facilities in Gran Canaria and Salamanca, which fortunately continues to grow today thanks to the tireless work and dedication of clinical veterinarians, whom we will never tire of thanking for their altruistic and selfless collaboration.
Answering your first concern (totally legitimate), these are not the same samples as the ones in the study you refer to (doi.org/10.3389/fvets.2020.564429). If this were the case, we would be making a methodological and praxis error, since obviously we should communicate it in the methodology and discussion of the study. Indeed, as the reviewer asks in the last sentence, we have collected more samples than those published in the aforementioned study. For example, some of the samples published in another paper that the reviewer also cites (10.3389/fvets.2021.784331) have also been collected partially coinciding with the time interval described in the paper doi.org/10.3389/fvets.2020.564429, being totally independent studies and samples.
Returning to your initial doubt, for this study we have collected samples from our sera collection taking into account the inclusion criteria of the study (>6 months of age, not having previous history of infection and not receiving chemoprophylaxis, and having the data of sex, age, habitat and postal code). With this, we have made a random and proportional selection for each province and autonomous community. Once selected, and as detailed in the methodology, the samples have been analyzed with the Uranotest Dirofilaria test kit.
We hope with this description to answer the reviewer's concerns: indeed, the funding sources are different as they are completely independent studies, and although Uranotest Quattro and Uranotest Dirofilaria are surely the same tests, we have only used the test described in the methodology. I think that with this we also respond to the doubts that arise with the results of Catalonia.
It is correct that all the authors of that previous research are the authors of this manuscript, and we also added new additions to the team of this publication. We are very proud of the latter, and we wish that the scientific system in Spain would not make it so difficult for us to keep such excellent young scientists with our group.
We hope that we have adequately clarified all the doubts of the reviewer; It seems to the authors very important that this matter be clarified, given the gravity of the doubts that call into question the prestige of our group.
According to table 1 six hundred forty seven blood samples were collected, and D. immitis was detected in 647 samples. It means 100 %.
We understand that the reviewer is referring to the results of Madrid. It is a typo that we have already corrected. Thanks for the observation.
Round 2
Reviewer 3 Report
The manuscript can be published in the present version.